# The TRPV6 Calcium Channel and Its Relationship with Cancer

**DOI:** 10.3390/biology13030168

**Published:** 2024-03-05

**Authors:** Yifang Wang, Xiaoling Deng, Rui Zhang, Hao Lyu, Shuai Xiao, Dong Guo, Declan William Ali, Marek Michalak, Cefan Zhou, Xing-Zhen Chen, Jingfeng Tang

**Affiliations:** 1National “111” Center for Cellular Regulation and Molecular Pharmaceutics, Key Laboratory of Fermentation Engineering (Ministry of Education), Hubei University of Technology, Wuhan 430068, China; 502000006@hbut.edu.cn (Y.W.); 502100011@hbut.edu.cn (X.D.); zhangrui1987@hbut.edu.cn (R.Z.); leohnyy@foxmail.com (H.L.); xiaoshuai825@hotmail.com (S.X.); jk1103@whu.edu.cn (D.G.); cefan@hbut.edu.cn (C.Z.); 2Cooperative Innovation Center of Industrial Fermentation (Ministry of Education & Hubei Province), Hubei Key Laboratory of Industrial Microbiology, Hubei University of Technology, Wuhan 430068, China; 3Department of Biological Sciences, University of Alberta, Edmonton, AB T6G 2E9, Canada; dali@ualberta.ca; 4Department of Biochemistry, University of Alberta, Edmonton, AB T6G 2H7, Canada; marek.michalak@ualberta.ca; 5Membrane Protein Disease Research Group, Department of Physiology, Faculty of Medicine and Dentistry, University of Alberta, Edmonton, AB T6G 2H7, Canada

**Keywords:** TRP channels, TRPV6, calcium signaling, cancer

## Abstract

**Simple Summary:**

TRPV6 is a highly selective channel for calcium ions and plays an important role in maintaining calcium homeostasis. Some evidence suggests that it is upregulated in advanced thyroid, ovarian, breast, colon, and prostate cancers. This review will summarize the function of TRPV6 in regulating calcium signaling in cancer and its potential application as a cancer therapeutic target.

**Abstract:**

Transient receptor potential vanilloid-6 (TRPV6) is a cation channel belonging to the TRP superfamily, specifically the vanilloid subfamily, and is the sixth member of this subfamily. Its presence in the body is primarily limited to the skin, ovaries, kidney, testes, and digestive tract epithelium. The body maintains calcium homeostasis using the TRPV6 channel, which has a greater calcium selectivity than the other TRP channels. Several pieces of evidence suggest that it is upregulated in the advanced stages of thyroid, ovarian, breast, colon, and prostate cancers. The function of TRPV6 in regulating calcium signaling in cancer will be covered in this review, along with its potential applications as a cancer treatment target.

## 1. Introduction

TRP channels are expressed in a wide range of cell types, including yeast and mammalian cells, and they make up a sizable and functionally varied superfamily [1]. The first TRP channel, called TRPC (transient receptor potential canonical), was identified in *Drosophila* and is involved in light perception [2,3]. On the basis of its sequence homology, the TRP superfamily can be subdivided into seven subfamilies: TRPC (canonical), TRPV (vanilloid), TRPM (melastatin), TRPP (polycystin), TRPML (mucolipin), TRPA (ankyrin), and TRPN (no mechanoreceptor potential C, or NOMPC) [4,5]. In vertebrates, six TRPV channels have been identified. TRPV1 mediates nociception and contributes to the detection and integration of diverse chemical and thermal stimuli. TRPV2 and TRPV3 are involved in thermal sensation, and are activated at temperatures in the warm and noxious heat ranges [6]. TRPV4 plays a role in osmosensing [7,8], nociception [9], and thermal sensing [7,8]. TRPV5 and TRPV6 are the most selective to Ca^2+^ in the TRP superfamily and play a role in Ca^2+^ reabsorption in the kidneys and intestines. TRPV5 and TRPV6 have P_Ca_/P_Na_ ratios over 100 and, thus, play essential roles in Ca^2+^-related intracellular pathways [10,11].

The most abundant metal element in the human body is calcium, which is found in all of the organs and tissues and is primarily found in the form of calcium carbonate. Many physiological functions are regulated by calcium, including the contraction of muscles and the heart, the transmission of neurotransmitters, the differentiation of tissues, and the metabolism of cells. Because the human body uses calcium for many physiological activities, it is critical for maintaining intracellular calcium homeostasis. The way the body obtains calcium is mainly through the daily diet. The human body’s ability to absorb calcium is essential for preserving calcium homeostasis, and abnormalities in intestinal ion absorption are typically linked to illness [12,13,14]. TRPV6 is thought to be a major epithelial calcium channel that absorbs calcium from the gut and many organs; therefore, TRPV6 is essential to human health [15,16].

Four identical subunits with six transmembrane segments each comprise the TRPV6 channel, which is an inward-recirculating Ca^2+^-selective ion channel [17,18]. TRPV6 is mainly distributed in the small intestine, epididymal epithelium, placenta, prostate, and exocrine pancreas. TRPV6 can promote Ca^2+^ uptake in the small intestine. TRPV6 maintains a low calcium level in semen in the epididymis and protects sperm from being killed [19]. Mice with a TRPV6 deletion have displayed low body weight, reduced fertility, and impaired intestinal Ca^2+^ absorption, among other symptoms. The abnormal transcriptional expression of TRPV6 has been shown to cause a series of diseases through the imbalance of calcium homeostasis [20]. The purpose of this review is to summarize the characteristics of the TRPV6 channels and their role in cancer, so as to determine the potential of TRPV6 in cancer treatments.

## 2. General Function and Structure of TRPV6

### 2.1. Structure of TRPV6

The functional structure of a human TRPV6 channel consists of four identical subunits arranged in a symmetrical tetramer. This structure includes two main compartments: a transmembrane (TM) domain with a central ion channel pore, and an intracellular skirt that is approximately 70 Å tall and 110 Å wide. The skirt is formed by the walls of the four subunits, enclosing a cavity that is approximately 50 Å by 50 Å and is located beneath the ion channel (Figure 1B) [21]. The N-terminal helix of a single TRPV6 subunit is followed by ankyrin repeat domains (ARDs), a linker domain containing a β-hairpin (composed of β-strands β1 and β2), a helix–turn–helix motif resembling the seventh ARD, and a pre-S1 helix connecting the linker domain to the TM domain [21]. Similar to the voltage-gated K^+^ or Na^+^ channels, TRPV6′s TM domain consists of six TM helices (S1–S6), as well as a pore loop (P-loop) that is located between S5 and S6. And the first four TM helices make up the S1–S4 domain [21]. In general, the S1–S4 domain serves as a voltage sensor for the voltage-gated ion channels. The movements of the positively charged arginine and lysine residues in S4 towards the membrane plane are influenced by the activation or deactivation of these channels [22]. However, in TRPV6, the S4 segment lacks positively charged residues. Nevertheless, the helical bundle conformation of the S1–S4 domain is rigidified by the presence of the many hydrophobic contacts between the aromatic side chains. This indicates that the domain remains rather immobile during the gating process [21]. After S6, the amphipathic TRP helix is aligned in parallel with the membrane. It functions as a central node where interactions between the hydrophilic and hydrophobic components of the intracellular skirt and TM domain occur [23] (Figure 1A).

Interactions between the subunits outside the membrane occur through contact between the S1–S2 extracellular loops and the S5-P and P-S6 loops of neighboring subunits. TRPV6 may be activated by β-glucuronidase klotho if the conserved N-linked glycosylation site (N358) in the S1–S2 loop is modified [24]. In the transmembrane (TM) domain, the S5, P-loop, and S6 pore module components engage in various intersubunit interactions that form the central ion channel pores (Figure 1C) [22]. Furthermore, the pore-forming components of every TRPV6 subunit establish a significant interaction with the S1 and S4 helices of the S1–S4 domain of an adjacent subunit. Consequently, the pore and S1–S4 domains engage in domain-swapped interactions with one another, akin to those of the voltage-gated K^+^ and Ca^2+^ channels and other TRP channels [22]. Channel-mediated calcium inflow is eliminated, however, when the S4–S5 linker is shortened in TRPV6*. This suggests that domain arrangement–switching mutations of various types can either improve or impair channel function [24,25]. The pathogenicity of the domain arrangement–switching mutations and the possibility of such remarkable domain architecture switches occurring in alternative ion channels have yet to be determined.

### 2.2. Ion-Conducting Pore

The TRPV6 pore structure comprises four essential components: the extracellular vestibule, selective filter, hydrophobic cavity, and lower portal. This structure spans from the exterior to the interior of the cell [22]. The P-loop helix connects to S5, and the S6 helix is the extracellular loop connecting this from the extracellular segment outside the TRPV6 channel. The inner segment of the cell is formed by TRPV6′s S6 helix, which also arranges ion conduction pathways. Each subunit region that joins S5 and S6 includes eight acidic residues. Four residue surfaces are responsible for conducting the ion channels and generating a strongly electronegative vestibule that confronts the exterior of the cell [21,23]. This vestibule is followed by a four-residue selective filter from the side chain of four highly conserved D541 residues (D542 in hTRPV6), one per protomer, to form a negatively charged ring or selective filter, which is particularly important for Ca^2+^ selectivity, permeability, and voltage-dependent Mg^2+^ block. The selective filter is a central cavity facing the hole. The pore characteristics of TRPV6 are highly selective for Ca^2+^ [26,27]. Studies have shown that if the key residue is D541, it will lose its selectivity for Ca^2+^ and prevent its absorption of Ca^2+^. The primary purpose of this is to generate a model that simulates the loss of TRPV6 function and to investigate the impact of this loss on animal physiology. Below the selected filter, a hydrophobic cavity is formed due to its enlarged aperture. Like a voltage-gated Na^+^ channel, this site permits the passage of small molecules and lipids [25]. The internal S6 spiral of the cell forms the lower gate of the hydrophobic cavity. The residue M577 in rTRPV6 (M578 in hTRPV6) forms a row into a narrow contraction point in the residue of the lower gate, which produces a hydrophobic effect, occluding the pores and keeping it in a closed state. In the Cryo-EM structure of human TRPV6, the lower gate residues facing the hole center are changed in the open states (N572 and I575) and the closed states (M578 and L574) [21,22,28].

### 2.3. TRPV6 Channel Gating Mechanism

McGoldrick et al. have recently unveiled the electron microscopy structure of human TRPV6 at low temperatures, both in its open and closed states (Figure 2). They observed that the side chains of N572, D542, I575, D580, and W583, along with the backbone carbonyl oxygen of I541, I540, and G579, are positioned on the surface of TRPV6 pores [21,22]. It can be concluded that the local transition of TRPV6′s S6 helix is indicative of the opening of the TRPV6 channels, since TRPV6 has an α-helix conformation when it is open and a π helix when it is closed. In the center of S6, there exists an α-to-π helical transition which modifies the orientation of the amino acid residue towards the central pore and, consequently, enlarges the diameter of the pore [21]. The residues facing the pore axis, for instance, are exchanged when the channel is closed and open, changing L574 and M578 into N572 and I575. Additionally, opening the lower door encourages the creation of a salt bridge between residues Q473, at the elbow of S4–S5, and R589, in the TRP helix. Moreover, it creates hydrogen bonds with T581, in S6, and D489, a residue in S5 [21]. However, the electrostatic interactions offset the adverse increase in energy resulting from the shift from an α helix to a π helix. The lower gate’s energy expenditures are the same in both the closed and open conformations; therefore, it is easily shifted between the two states by a variety of physiological stimuli [21].

### 2.4. Regulation by Calmodulin (CaM) and Phosphatidylinositol 4,5-Bisphosphate (PIP2)

The relationship between PIP2 and Ca^2+^-CaM is linked to the regulation of TRPV6 channel activity (Figure 3). Two negative feedback mechanisms are activated following the intracellular Ca^2+^ inflow. These mechanisms primarily involve TRPV6 channel inactivation mediated by CaM binding and PIP2 depletion, which can lower TRPV6 channel activity and prevent intracellular Ca^2+^ overload [29]. PIP2 can activate most of the TRP channels, including TRPV5 and TRPV6. The pathophysiology of multiple disorders, including Dent’s disease, Lowe’s syndrome, and cystic fibrosis, has been linked to PIP2-mediated control of TRPV6. There is an ongoing debate regarding PIP2′s binding site to TRPV6 [25]. In a more recent study, Cai et al. discovered that PIP2 inhibits the self-inhibition process of the intramolecular contacts of TRPV6′s S4–S5 connector by binding to three cationic residues at the S5 or C terminus [30]. This activates the TRPV6 channel function. Niemeyer et al. made the initial discovery that the binding site between CaM and TRPV6 is predominantly situated in the protein’s C-terminal region. Likewise, they discovered that this area also contributes to TRPV6′s Ca^2+^-dependent deactivation mechanism [31]. Research has indicated that TRPV6 activity can be inhibited by the distal C region of TRPV6 binding to the Ca^2+^-CaM complex. TRPV6-CaM complexes are stoichiometric at 1:1; moreover, the fully Ca^2+^ bound N-terminal and C-terminal lobes of CaM can adopt a unique head-to-tail arrangement. Within a cell, the more TRPV6 binds to CaM, the more dynamic [Ca^2+^] becomes [23]. How does PIP2 affect Ca^2+^-CaM-mediated TRPV6 inhibition? Ca^2+^-CaM-mediated TRPV6 inhibition is dependent on the PIP2 concentration, and an equilibrium between the intracellular Ca^2+^ and PIP2 concentrations controls the CaM inactivation of TRPV6 [32]. How does a Ca^2+^ influx affect CaM-mediated TRPV6 channel inhibition? First, the N-terminus and C-terminal of CaM are easy to bind to Ca^2+^ and, under this condition, when Ca^2+^ perpetually binds to the C-terminal of CaM, it binds to the distal C-terminus of TRPV6. A significant amount of Ca^2+^ will enter while the TRPV6 channel is active. When the combination of CaM and Ca^2+^ approaches saturation, it blocks the entry of ions from the pore and inhibits the TRPV6 channel’s activity [28,33,34].

## 3. TRPV6 Expression in Cancer

Recent research has suggested that TRPV6 is an oncochannel and that its gene is an oncogene; nonetheless, it is more likely to be a proto-oncogene [35]. Consequently, prostate, colon, breast, thyroid, and ovary carcinomas exhibit much higher levels of TRPV6 channels in comparison to normal tissues or cells [36,37].

### 3.1. Calcium and Cancer

Multiple illnesses are linked to Ca^2+^ homeostasis, and modifications to this state impact each stage of the metastatic process, such as angiogenesis, invasive migration, and EMT [38]. The calcium channel embedded within the serous membrane plays a crucial role in regulating the balance of Ca^2+^ ions in the body. Mechanosensitive ion channels, which enable the activation of downstream pathways when Ca^2+^ comes in, may be responsible for mediating many of the physical characteristics that have a beneficial effect for the growth of cancer [39,40]. The Ca^2+^ triggering mechanism, which involves the nuclear factor Kappa light-chain enhancer, and which activates the B-cell (NF-κB) pathway and calcium/calmodulin-dependent kinase II (CaMKII), controls apoptosis and the cycle [41,42]. The NF-κB protein complex regulates DNA transcription in response to various stimuli, and has a role in inflammation, the immunological response, proliferation regulation, and cell survival [43]. Other important calcium-dependent factors are calcium-dependent cysteine protease, calpain, and calcium-dependent serine–threonine phosphatase, and calcitonin, which control the cell cycle, apoptosis, and cell migration [39]. Consequently, an aberrant mechanosensitive Ca^2+^ channel expression by these or other, less well-studied signaling cascades may be crucial for the development of cancer. Furthermore, it has been observed that the expression of these channels alters as a tumor grows [39].

TRPV6^−/−^knockout mice (TRPV6^−/−^) have been found to have substantial impairments in male fertility, increased urine Ca^2+^ excretion, reduced femur density, weight loss, dermatitis, and abnormalities in intestinal Ca^2+^ absorption [44]. Recently, a variety of human TRPV6 variants have been associated with transient neonatal hyperparathyroidism, undermineralization, and fetal skeletal dysplasia. Because calcium is so important in the formation of cancer, TRPV6 has also been classified as a tumor channel implicated in enhanced cell proliferation and the prevention of apoptosis. This protein is overexpressed in various human cancer types, including leukemia, breast, prostate, colon, ovarian, thyroid, and endometrial malignancies (Figure 4) [19,45]. Consequently, TRPV6 inhibitors have the potential to treat a wide range of illnesses, including TRPV6-rich cancers, that are linked to abnormalities in calcium homeostasis [46].

### 3.2. TRPV6 as an Oncochannel and its Mechanism of Action

Due to reports that TRPV6′s mRNA and protein are overexpressed in several human cancers [45], the gene for TRPV6 has been classified as an oncogene [47,48], and the channel as an oncochannel [49]. On the other hand, there is no proof that proto-oncogenes or cancer can be caused by TRPV6 expression alone [35]. The only certain thing is that TRPV6 has a direct role in the calcium-dependent growth of cancer cells, while its precise function in this process is unknown [50]. Different nuclear factors of activated T cells (NFAT) are activated in prostate and breast cancer cells by TRPV6 activity and the cytosolic calcium levels that follow [51,52]. TRPV6 knockdown, in the research, has resulted in enhanced apoptosis and decreased proliferation. High amounts of intracellular calcium are sustained when TRPV6, which is constitutively active under physiological circumstances, is overexpressed [19]. The activation of the calcineurin/calmodulin/NFAT pathway requires this calcium. A dephosphorylated NFAT in the nucleus has a short half-life (≤20 min), hence, maintaining cytosolic calcium levels is necessary to support the cell survival response. The different sets of auxiliary transcription factors (TFs) and other regulatory proteins that may interact with NFATs are what determine this cell response [53,54]. This pathway originates from internal signals, such as endoplasmic reticulum emptying through IP3-receptor activation [55]. The activation of Orai1 channels on the plasma membrane is triggered by the conformational changes in the calcium sensor STIM1. This occurs when the calcium levels in the endoplasmic reticulum decrease [55]. When these Orai1 channels open, the intracellular Ca^2+^ stores, like those of the endoplasmic reticulum, can be replenished with Ca^2+^ [35].

Calcineurin, a phosphatase that is driven by Ca^2+^/calmodulin, specifically acts on an NFAT, a transcription factor that is hyperphosphorylated. The activation of an NFAT occurs through its dephosphorylation [56]. Apoptosis is a major contributing factor to the development of cancer, and NFATs are a prevalent regulator of the cell cycle [57]. A thorough evaluation of the function of NFATs in the initiation and progression of cancer has been conducted [56]. Upon its activation, an NFAT translocates to the nucleus and engages in interactions with Jun/Fos and other proteins to modulate gene expression [57]. Thus, matrix metalloproteinase types 1 and 2, as well as autotaxin, are examples of active genes that are linked to proliferation and migration [58]. The phospholipase autotaxin is released and generates lysophosphatidic choline, which binds to the lysophosphatidic acid receptor 1 [59]. As a result of the upregulating of the anti-apoptotic protein Bcl-2, which is known to limit the release of cytochrome c from the mitochondria, and so, stop the development of apoptosomal processes, TRPV6 may have these well-known anti-apoptotic effects [60].

The elevated levels of calcium due to TRPV6 may also lead to the accumulation of hydroxyapatite crystals, reported in some tumors, which may upregulate matrix metalloproteinases (MMPs) [61]. Numerous genes, as well as the dozen additional proteins that NFATs can form transcription complexes with, are regulated by the four known calcineurin-dependent isoforms of NFATs [57]. The Orai1 protein of the store-operated calcium channel system is necessary for TRPV6′s enhanced trafficking to the plasma membrane, which plays a role in prostate cancer [35]. Together with increased TRPV6 trafficking, cell proliferation is increased, and basal apoptosis is reduced, with a greater resistance to cisplatin and thapsigargin. Thus, TRPV6 activity is associated with the survival response of several prostate cancer cell lines (LNCaP, PC-3, and DU145). The data suggest that cancer cells have a greater sensitivity to and need for sustained intracellular calcium than do normal tissues [35].

### 3.3. Prostate Cancer

The most prevalent noncutaneous cancer in humans, and the second most lethal cancer in men is still prostate cancer, which is most prevalent in industrialized countries [62]. According to recent research, TRPV6 transcripts are not found in normal prostate tissue, but are expressed in metastatic prostate cancer, locally progressed prostate adenocarcinoma, and androgen-insensitive prostate lesions. TRPV6 has become a promising prognostic marker in recent years [63,64]. In contrast to its close association with the Gleason >7 grade (Gleason score: a system for evaluating the malignancy of prostate cancer, and a score of more than seven is considered a medium-risk prostate cancer), TRPV6 is not linked to the development of prostate cancer, although it is abundant in advanced prostate cancer. Instead, it serves as a powerful indicator of the advancement of a tumor and its subsequent infiltration into healthy organs [63,65,66]. Previous research has demonstrated that TRPV6 is involved in high calcium-selective currents in prostate cells and is tightly regulated by intracellular Ca^2+^ concentrations. [66]. Multiple papers have indicated that TRPV6 plays a direct role in regulating the growth of prostate cancer cells (specifically the LNCaP cell line). It achieves this by reducing the rate of cell proliferation, causing cells to accumulate in the S-phase of the cell cycle, and raising the expression of proliferating cell nuclear antigen (PCNA). Moreover, TRPV6 facilitates the uptake of Ca^2+^ into LNCaP cells, subsequently activating NFAT downstream [51,67,68]. Androgen receptors control TRPV6 expression in LNCaP cells, although this regulation is not ligand independent [51,69]. The androgen receptor agonist dihydrotestosterone inhibits TRPV6 expression, whereas the androgen receptor antagonist bicalutamide enhances TRPV6 expression [70]. Because of this, this channel offers a potential therapeutic target for promoting the advancement of novel strategies for the treatment of prostate cancer. In short, prostate cancer cells may be able to sustain a greater rate of proliferation, higher cell survival, and resistance to apoptosis because of the overexpression of the TRPV6 Ca^2+^ channel.

### 3.4. Breast Cancer

The most frequent and fatal cancer in women worldwide is breast cancer [71]. Triple-negative breast cancer, also known as basal-like breast cancer, is the deadliest type of the disease and currently has the least effective treatment. According to earlier research, breast tumor tissue expresses more TRPV6 mRNA and protein than normal breast tissue. The TRPV6 protein was discovered to be prevalent in the breast tumor tissue invading region by further investigations [52,72,73].

Cancer cells can be prevented from migrating, invading, or surviving by the expression of TRPV6 knockdown, or by MCF-7, MDA-MB-231, and T-47D cell line inhibitors. Furthermore, in breast cancer cell lines with increased endogenous TRPV6 expression, silencing TRPV6 expression resulted in decreased basal calcium inflow and cell proliferation, along with decreased DNA synthesis [71,72]. When TRPV6 was silenced using siRNA in the breast T-47D cell line, there was an increase in apoptosis and a decrease in cell growth [74]. TRPV6 is upregulated in metastatic breast cancer, and its overexpression or upregulation accelerates the migration of the primary breast cancer cells. Conversely, TRPV6 inhibition reduces cell migration [52]. Cai et al. found that the TRPV6 pathogenic mutation R532Q enhances the TRPV6/PI3K interaction, thereby activating the PI3K/Akt/GSK-3β cascade to further promote breast cancer migration [75]. TRPV6 has a strong effect on the proliferation of breast cancer cells, which are regulated by estrogen, progesterone, tamoxifen, and 1, 25-vitamin D3. When TRPV6 expression is inhibited by small interfering RNA, the effect of tamoxifen on cell viability is enhanced [52]. This suggests that TRPV6 may be a new target for the development of calcium channel inhibitors for the treatment of breast adenocarcinoma expressing TRPV6 [52]. The strong correlation between TRPV6 expression levels and the levels of common EMT markers implies that TRPV6 may have a function in the mesenchymal invasion of breast cancer cells [76]. However, it remains uncertain if the tumor transformation of mammary epithelial cells is linked to the dysfunction of this TRPV6 regulatory component.

### 3.5. Ovarian Cancer

An early report that revealed an over-expressing TRPV6 channel in one biopsy report also mentioned ovarian cancer [77]. According to a recent study, ovarian cancers of various grades—low-grade serous, high-grade serous, clear cell, endometrioid, and mucous—express more TRPV6 mRNA than normal tissues do [78]. All five ovarian cancer disease types have been shown to have elevated TRPV6 mRNA levels in both their early and late stages when compared to healthy tissues [78]. The research suggests that a plausible working model for the mechanism of action of elevated TRPV6 is as follows: sustained elevated cytosolic calcium, as a result of the amounts or activity of TRPV6 increase, binds to CaM, which then activates calcineurin, a CaM/Ca^2+^ activated phosphatase [79]. Following its translocation to the nucleus, the hyper-phosphorylated NFAT transcription factor, which is triggered by activated calcineurin, can subsequently activate the genes that affect cell migration and proliferation [78]. For membrane type 1 matrix metalloproteinase, matrix metalloproteinase-type 2, and secreted autotaxin, the NFAT functions as a transcription factor [58]. The latter generates a Growth Factor Receptor (GFR)-activating extracellular lysophosphatidyl choline. An elevated TRPV6 function has an anti-apoptotic effect because it increases the production of anti-apoptotic Bcl-2. This protein blocks the release of mitochondrial cytochrome c and prevents the development of apoptosomes. The increase in Bcl-2 production is reliant on the NFAT [80]. Reduced cell proliferation and higher cell death are the results of RNA silencing, which diminish TRPV6 synthesis in cancer cell lines. This is likely due to the NFAT signaling pathway reverting at lower intracellular calcium levels [80]. Whether the NFAT/calcin pathway and downstream anti-apoptotic events are active in ovarian cancer is unknown, but it provides an idea for further research. Jiang et al. found that low concentration of lidocaine reduced the invasion and migration ability of ovarian cancer ES-2 cells expressing TRPV6 [81]. Furthermore, SKOV-3 tumor xenografts in mice were shown to develop less when TRPV6 was targeted using TRPV6 peptide antagonists, supporting the idea that TRPV6 may be a key target for ovarian cancer treatments [78].

### 3.6. Pancreatic Cancer

The presence of TRPV6 is strongly associated with the progression and outcome of pancreatic cancer (PC), as patients with elevated levels of TRPV6 in their tumors have a poorer chance of survival [37]. In pancreatic cell lines, an siRNA-mediated TRPV6 knockdown reduced invasion and proliferation, while inducing apoptosis and cell cycle arrest. TRPV6 plays a promising role in the development and progression of PC [82]. TRPV6 has been discovered to have Numb as an interacting partner [66]. The activity of TRPV6 in breast cancer cells is suppressed by Numb through electrostatic contact, which controls the influx of Ca^2+^ [83,84]. The GSK3β, AKT, and MAP kinase pathways implicated in TRPV6-regulated cell proliferation are activated by a Ca^2+^ influx. In prostate cancer cell lines, silencing Numb decreased TRPV6 expression [67,84]. In contrast, TPRV6 expression was elevated in breast cancer cells with a Numb knockdown. TRPV6 and Numb control each other’s protein breakdown and stability. Therefore, TRPV6/Numb enhances TRPV6′s potential as a PC therapeutic target [82].

### 3.7. Colon Cancer

TRPV6 channel overexpression has been linked to early-stage colon cancer, while the suppression of TRPV6 has hindered cell growth and triggered programmed cell death in colon carcinoma cells [19]. It has been demonstrated that both human colon cancer cell proliferation and colonic crypt hyperplasia in mice are influenced by abnormal TRPV6 expression. Consequently, TRPV6 contributes to gastrointestinal malignancies, particularly in their first phases [85]. In late-stage malignancies (stages III and IV), TRPV6 was either not detected at all or was present at extremely low levels, whereas TRPV6 overexpression was detected in 66% of stage I tumors and 17% of stage II tumors. TRPV6 has been shown to be highly expressed in a colorectal cancer cell line (SW480), indicating its significant therapeutic relevance [86]. Dietary calcium has been shown to reduce the risk of colon cancer, and Ca^2+^ intake in a high-calcium diet may require the inhibition of TRPV6 to have a protective effect on the colon [19]. While the exact mechanism of action of TRPV6 in colon cancer remains unknown, new research has shown that invasive qualities appear to be conferred on colon cancer cells by changes in TRPV6′s binding structure to CaM [86]. In contrast, it has been observed that the expression of TRPV6 in colon cancer cells is elevated due to enhanced vitamin D signaling through p38α and GADD45α in the colorectal cell line SW480 [87]. TRPV6 regulates colon cancer by enhancing the IGF-induced PI3K-PDK1-Akt signaling pathway in human colon cancer [88].

## 4. Mechanism of TRPV6 Channel Expression Control in Cancer

The fact that several epithelial-type malignancies (over)express TRPV6 at both the mRNA and protein levels in comparison to healthy tissues is now widely recognized. However, the mechanism responsible for the overexpression and/or overregulation of the TRPV6 gene (7q33-34) is still mostly unknown [45].

### 4.1. TRPV6 Transcription Control by Vitamin D3 Receptor (VDR)

The VDR was shown to bind to the response element in the TRPV6 gene and, thus, activate its transcription [89]. Nonetheless, vitamin D3′s transcriptional regulation of TRPV6 is intricate. Vitamin D3 not only induces TRPV6 expression, but also stimulates the transcription of growth arrest and GADD45α, a protein that is generated in response to stress or DNA damage [90]. GADD45α has been shown to activate mitogen-activated protein 3 kinase 4 (MEKK4), which mediates activation of both p38α and c-Jun N-terminal kinase (JNK) [91]. P38α is a 38-kD protein kinase that is typically activated through phosphorylation and is produced in response to cellular stress. In addition to its role in vitamin D3 activation, p38α also stimulates the transcription of TRPV6, which is dependent on vitamin D3 [91]. However, p38 has also been observed to stimulate the proliferation of several cancer cell lines, including HeLa, prostate, breast, melanoma, and chondrosarcoma [92]. Thus, upregulating TRPV6 is the way that vitamin D3 modulates cancer.

### 4.2. TRPV6 Transcription Control by Androgen Receptor (AR)

The role of the AR in the regulation of TRPV6 was initially investigated by studying the effects of receptor agonists and antagonists. It was observed that TRPV6 expression was increased by an AR antagonist (bicalutamide), but inhibited by an AR agonist (dihydrotestosterone) [93]. The androgen-sensitive prostate cell line LNCaP was used to study the mechanism by which the AR regulates TRPV6 expression [94]. The AR was found to be ligand-independently sensitive to TRPV6 expression, indicating that the AR functions as a co-regulator rather than a direct regulator of TRPV6 transcription [94]. The NFAT system receives a downstream signal from increased calcium concentration, and when the AR is knocked down with siRNA, the TRPV6 mRNA and protein are reduced by 48 and 72 h, respectively, after treatment.

### 4.3. TRPV6 Transcriptional Control by Estrogen Receptor

The TRPV6 gene’s promoter contains an estrogen receptor-responsive region. In the breast cancer cell line T-47D, tamoxifen [95], an estrogen receptor antagonist, caused TRPV6 mRNA to be downregulated, whereas progesterone and estradiol elevated it [95]. Furthermore, it has been discovered that TRPV6 increases in breast and prostate cancers in an estrogen-dependent manner. This upregulation may function in tandem with other transcription factors that are stimulated by increased calcium levels [19].

### 4.4. TRPV6 Transcriptional Control by PPARα

The presence of a response element in the TRPV6 gene suggests that peroxisomal proliferator-activated receptor alpha (PPARα), a nuclear receptor, may influence TRPV6. However, the specific influence of PPARα ligands, such as fibrates, endocannabinoids, and polyunsaturated fatty acids, on the state of TRPV6 is still unknown [94]. There may be a potential connection between PPARα, endocannabinoids like anandamide, and TRPV6 under conditions of oxidative stress. There may be connections between retinoids and fatty acids to cancer, as both PPARα and vitamin D need the formation of a dimer with retinoic acid receptor α [95].

### 4.5. TRPV6 Transcriptional Control by Other Transcriptional Factors

The TRPV6 gene’s promoter region contains a large number of transcription factor (TF) binding sites [96]. The following likely TFs are listed in the database: HOXA5 MAZ, NKX2-1, PPARA, TLX2, and ZEB1. TRPV6 expression is controlled by transcription factors GATA1, GLI2, HNF1A, KLF13, MTF1, NFE2, NR5A2, RBPJ, and VDR, according to known binding site sequences [96]. Interestingly, the literature does not provide evidence that any of the listed transcription factors have an impact on TRPV6 expression, except for VDR and PPARα. On the other hand, TRPV6 is known to be regulated by the AR and estrogen receptors, which are absent from the database entry [97]. Every one of the TFs mentioned above has a role in one or more of the oncogenic processes [97].

## 5. Pharmacology of TRPV6

Due to TRPV6′s critical function in the development of cancer, it is imperative to identify small molecule inhibitors and TRPV6 ion channel blockers as possible therapeutic targets. These mainly include PCHPDs, RR, 2-APB, Econazole, THCV, SOR-C13 synthetic peptide, capsaicin, Cis-22a, and TH-1177(33,95). PCHPDs and RR are not directly blocked by blocking ion channel pores, Econazole and 2-APB are isomerized by the accompanying lipid shift, and THCV may also belong to the second category [98]. Prostate and ovarian cancer growth have been demonstrated to be inhibited in cells and animal models by the 13-amino acid peptide SOR-C13 in completed phase I clinical studies. At the moment, there are very few small molecule inhibitors of TRPV6 that have passed clinical trials and are approved drugs [77,78,99]. The TRPV6 inhibitors are summarized in Table 1.

## 6. Conclusions and Perspectives

The epithelial Ca^2+^-selective channel TRPV6 has drawn interest due to its possible involvement in the development of cancer. One important messenger that controls a variety of signaling pathways related to cellular functions is Ca^2+^. It is commonly known that a significant portion of the hallmark processes in the course of cancer are caused by calcium signaling mechanisms, which are also deeply involved in the regulation of cell proliferation and death. Furthermore, TRPV6 is a strong target that can be used to further disturb the aberrant calcium homeostasis that is needed and seen in a lot of malignancies. It has been shown that decreasing TRPV6 activity by medication-induced inhibition or by lowering its expression is useful in four types of cancer: pancreatic, ovarian, breast, and prostate. Over the past few years, xenografted animal models and cancer cell lines have shown evidence that TRPV6 inhibits solid tumors. It is now also effective in humans. Numerous investigations have examined TRPV6′s regulatory mechanism and the medications that can be used to block its expression; the discovery of SOR-C13 has allowed for the exploration of new therapeutic targets for malignancies linked to TRPV6. TRPV6, therefore, has importance and value in clinique and can be employed as a marker for tumor identification and prognostic evaluation, as well as a target for disease treatment.

## Figures and Tables

**Figure 1 biology-13-00168-f001:**
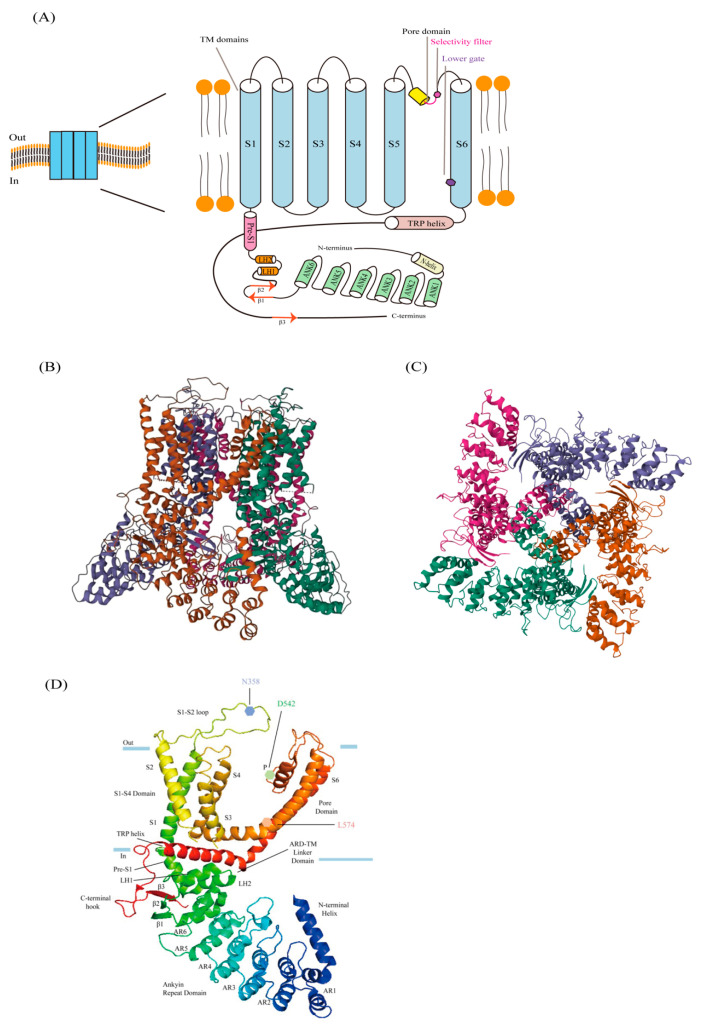
(**A**) The human TRPV6 domain organization. The N-terminal helix (light yellow), the ankyrin repeat domain with six ankyrin repeats (green, ANK 1-6), the binding domain composed of the β-hairpin (β1 and β2, in red), and the two linker helices (orange, LH1 and LH2) are the secondary structural features found in the TRPV6 monomer subunit. The transmembrane (TM) domain, which consists of six TM helices (S1–S6, sky blue), the amphiphilic TRP helix, the β strand that forms a β-sheet with β1 and β2 (red, β3), and the pore helix that joins S5 and S6, are all connected by the front S1 helix (pink). (**B**,**C**) The human TRPV6 tetramer is shown from the side and the top (PDB ID: 6BO8). (**D**) A glycosylation site (N358), a key selective residue in the selective filter (D542), and a representative residue in the lower gate (L574), are identified within a single human TRPV6 subunit.

**Figure 2 biology-13-00168-f002:**
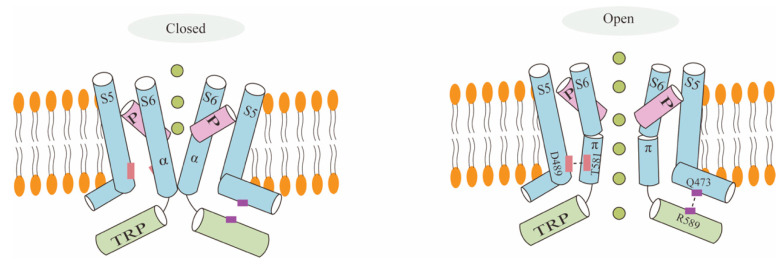
The cartoon represents the closed and open states of TRPV6. TRPV6 undergoes a local α-π spiral transition in S6, together with its transition from the closed to the open state, by controlling the lower gate’s conformational change and by an electrostatic bond’s creation without changing the idea of the selectivity filter. A salt bridge forms between Q473-R589, hydrogen bonds form between D489-T581, the pore size widens, Ca^2+^ is allowed to pass through the lower gate, and the open state is stabilized by the formation of electrostatic interactions, which may counteract the energy cost of the unfavorable α-to-π spiral transition. When the bottom door is closed, its remnants will constrict the opening and create a hydrophobic sealing.

**Figure 3 biology-13-00168-f003:**
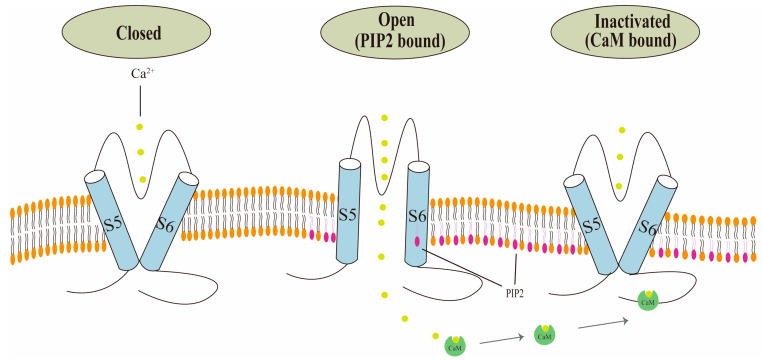
TRPV6 channels become inactivated when PIP2 binds to them, but they become activated when PIP2 binds to CaM. Conformational alterations cause TRPV6 to become activated when PIP2 binds to it. Additional Ca^2+^ ions attach to CaM when internal Ca^2+^ is present, creating a Ca^2+^-CaM complex that inactivates the channel.

**Figure 4 biology-13-00168-f004:**
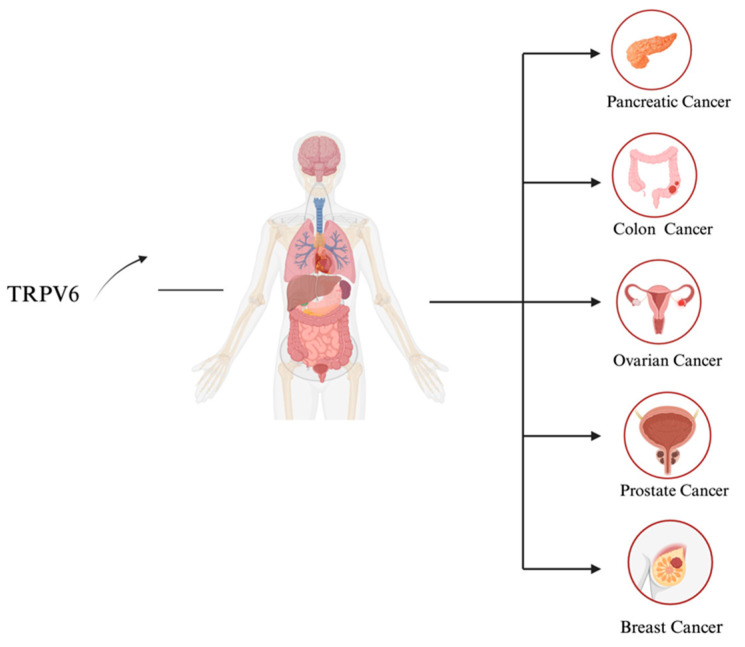
Overexpression of TRPV6 is associated with development of multiple cancers. Created with BioRender.com.

**Table 1 biology-13-00168-t001:** Summary of compounds that inhibit TRPV6.

Compound	Remarks
PCHDs	Highly selective TRPV6 inhibitors [100]. Inhibit TRPV6 by mimicking CaM and blocking ion channel pores [21,33]. Unique inactivation simulation mechanism [98].
Ruthenium Red (RR)	Inhibits most TRP channels [101]. Blocking ion channel pores inhibit TRPV6 [102]. Inhibiting Ca^2+^ from cytoplasm into mitochondria maintains Ca^2+^ homeostasis [103].
2-aminoethoxydiphenyl borate (2-APB)	Mainly combined with base points S1–S4 of TRPV6 [33]. Isomerization inhibition of TRPV6 lipid shift [33]. Competitive antagonist of IP3R and regulator of intracellular Ca^2+^ concentration [104].
Econazole	FDA-approved antifungal agent and TRPV6 antagonist [105]. Main inhibition site is the S4–S5 interface [98]. Ca^2+^ channel antagonist and inhibitor of cancer cell invasion [106].
Phytocannabinoid tetrahydrocannabivarin (THCV)	Binding in the transmembrane region of TRPV6 [98]. Natural cannabinoids extracted from cannabis, non-psychoactive analogues [107]. Regulate TRPV6-dependent epithelial calcium transport [107].
SOR-C13 (13 amino acid peptide)	Tumor inhibition was effective in ovarian and breast cancer xenotransplantation models [79]. Potential of imaging to diagnose cancer in vivo [80]. The paralyzing venom of the northern short-tailed sorex (Blarina brevicauda) has a peptide consisting of 54 amino acids [80].
TH-1177	Inhibit low-pressure activation of T-type Ca (2+) channels (TCC) [108]. Blocking calcium entry channels in prostate cancer cells both in vitro and in vivo also increased the average lifespan of mice with prostate tumors [108]. Nine related compounds [108].
Capsaicin	Apoptosis of human gastric cancer cells is induced by TRPV6 pathway [109,110].
Piperazine derivative Cis-22a	The first submicromolar, small molecule inhibitor, it decreased the cell viability of tumor cell lines overexpressing TRPV6 [111].

## Data Availability

Not applicable.

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
