# Peer review of "The TRPV6 Calcium Channel and Its Relationship with Cancer"

_biology, 2024, doi:10.3390/biology13030168_

Round 1
Reviewer 1 Report
Comments and Suggestions for Authors
please see attached file

I would suggest that the authors have the manuscript revised by a native speaker.
Reviewer 2 Report
Comments and Suggestions for Authors
This is a straightforward review of a topic of current interest. The manuscript does focus on the material described in the title and abstract. The manuscript reds reasonably well. However, there are a number of editorial problems.
Introduction:
The uncited statement that calcium is the most common in in the body is a ludicrous statement.
The body gets its calcium from the diet, not from "dietary requirements".
Main text:
The parts of Figure 1 are not labelled A, B, and C in the figure itself. The source of the data used to generate the figures should be cited in the Figure caption.
I could not find the abbreviation for calmodulin was ever identified. The abbreviation also varies between "CaM" and "cam". Subsequently, I found numerous abbreviations that were never defined. A reviewer unfamiliar with the field would find much of the last half of the manuscript uninterpretable.
The charges on ions should be superscripts.
Overall, this could be a good review article; however, the authors need to be much more careful in editing.
Comments on the Quality of English LanguageThe English grammar is good enough not to distract the reader. However, some finer points of English grammar are occasionally incorrect. Verb tense changes within some paragraphs, for example. Font size changes in some places in the manuscript.
How recent is "recent"? Some research described as recent is a decade old.
Round 2
Reviewer 2 Report
Comments and Suggestions for Authors
revisions are satisfactory for publication
